# Accumulation of road salt in a calcareous fen: Kampoosa Bog, western Massachusetts

**Wayne Ndlovu**[1,2,3]\*, **Andrew J. Guswa**[4], **Amy L. Rhodes**[3]

**1** Kansas Geological Survey, University of Kansas, Lawrence, Kansas, United States of America,
**2** Department of Geology, University of Kansas, Lawrence, Kansas, United States of America, **3** Department of Geosciences, Smith College, Northampton, Massachusetts, United States of America, **4** Picker Engineering Program, Smith College, Northampton, Massachusetts, United States of America

\* wndlovu@ku.edu

**Data Availability Statement:** All relevant data for this study are publicly available from the HydroShare repository (https://doi.org/10.4211/hs.2c676a5913f44c85befedfcedc6185bc).

## Abstract

Road salt poses a threat to the quality of soils and water resources. Wetlands located in salt contaminated areas are at risk of experiencing lower plant and animal species diversity. Therefore, it is critical to understand how modifications to salt application rates and hydrological events impact wetland water quality. Here, we use chloride mass flux, discharge, groundwater chloride concentration, meteorological, and salt application data from 2012–2020 to estimate chloride accumulation and outflux rates in the Kampoosa Bog subwatersheds, located in Stockbridge and Lee, Massachusetts, and bordered by major highways (Interstate-90 and U.S. Route 7). We also investigate the correlation between wetland size and chloride retention rate. During the 2018–2019 period, mean annual chloride application rates in the major watershed increased from 363000 kg/year (2012–2017) to 479000 kg/year. This led to a net chloride accumulation (KB100 subwatershed: 339000 kg; KB150 subwatershed: 188000 kg) and increased groundwater chloride concentrations in the fen. Chloride outflux from these subwatersheds was primarily driven by discharge. We found that the relationship between wetland percent cover and chloride retention is complex. Although the percent wetland cover is greater in the KB100 main wetland region compared to the KB150 subwatershed, high precipitation in 2018 resulted in similar chloride retention efficiencies (~26%). During the drier year (2019), chloride retention was higher in the wetland region due to its gentle slopes which promote water accumulation and consequently higher evaporation rates which lowers discharge and chloride outfluxes. The chloride steady-state concentration analysis also suggests that there is potential for chloride accumulation to continue because the watershed has not yet reached steady-state chloride concentrations. Without major modifications to salting practices, chloride concentrations will continue increasing and potentially promote the re-growth of invasives (*Phragmites*) and continued growth of salt tolerant species (*Typha angustifolia/xglauca*) that diminish plant diversity.

## Introduction

Concerns over road-salt application rates have grown over the past few years. Annually, large quantities of road salt are applied to roads and other impervious surfaces in the snowy

**Funding:** Wayne Ndlovu reports financial support was provided by the Natural Heritage and Endangered Species Program of the Massachusetts Division of Fisheries & Wildlife in the Department of Fish and Game (RFR #DFW-2020-051), McKinley Fellowship and the Tomlinson Fund (Smith College). Amy L. Rhodes reports financial support was provided by Natural Heritage and Endangered Species Program of the Massachusetts Division of Fisheries & Wildlife in the Department of Fish and Game (RFR #DFW-2020-051). If there are other authors, they declare that they have no known competing financial interests or personal relationships that could have appeared to influence the work reported in this paper. The funders had no role in study design, data collection and analysis, decision to publish, or preparation of the manuscript.

**Competing interests:** The authors declare that they have no known competing financial interests or personal relationships that could have appeared to influence the work reported in this paper.

northeastern and midwestern USA to reduce vehicle accidents by about 78–87% [1, 2] and to increase road safety [3]. During the snow and ice months, ice formation reduces pavement friction [4, 5] and road-salts act as anti-icing agents which prevent ice from bonding to pavements or as deicing agents that lower the freezing point of water [6–9]. In recent decades, salt application on USA highways has increased; from less than 5000 tonnes/year in the early 1940s to over 20 million tonnes/year in 2010 due to increases in impervious surfaces, road densities and population densities [10].

Several chloride-based salts like sodium chloride (NaCl), calcium chloride ($CaCl_2$) and magnesium chloride ($MgCl_2$), and organic salts like calcium magnesium acetate (CMA) and potassium formate (KCOOH) [5, 11] are used as deicers, and NaCl is used most commonly due to its abundance and affordability [6]. In particular, NaCl makes up more than 55% of chloride-based salts used in the USA and Canada [12]. Despite its benefits, NaCl increases water salinity by combining with precipitation and surface runoff to form saline water which can contaminate groundwater and streamflow [13–15].

High Na and Cl concentrations pose a risk to human health by mobilizing heavy metals bound to soils and can potentially increase the concentration of dissolved metals in drinking water [16–18]. Chloride also corrodes water infrastructure causing high chloride to sulfate mass ratios (CSMR) and lead leaching [19, 20]. Additionally, road-salt can directly contaminate drinking water by increasing chloride concentrations to levels above the U.S. Environmental Protection Agency (EPA) secondary maximum contaminant level standard of 250 mg/L [21]. In some parts of the USA, chloride concentrations above 250 mg/L threshold have already been detected in streams located in urban watersheds (>1500 mg/L in Milwaukee metropolitan area in 2007; [22], >1000 mg/L in the Minneapolis/St Paul Twin Cities Metropolitan Area; [14], and >10000 mg/L in the Mohawk River Basin in the 1990s; [23]).

Increased salinization also leads to physical, chemical, and biological alterations of the environment [15]. Wetland soils located in salt contaminated watersheds tend to have elevated sodium levels because they adsorb $Na^+$ during cation exchange [24–26]. On average, chloride concentrations in freshwater wetlands located in forested areas that are mainly recharged by groundwater are < 3 mg/L– 100 mg/L, but higher (> 200 mg/L) concentrations have been recorded in areas closer to road networks [25–28]. High salinity in wetlands has been linked to changes in plant species diversity and distribution; salt-tolerant species may grow at a faster rate and inhibit growth of other species [26, 29]. For example, Cañedo-Argüelles [30] and Szklarek et al. [29] showed that 5% of aquatic species exposed to saline water are affected by chloride concentrations >210 mg/L while 10% are affected by concentrations >240 mg/L. Saline wetlands are usually characterized by cattail (*Typha latifolia*), common reed (*Phragmites australis*) and red maple (*Acer rubrum*) which are salt tolerant [25, 26, 31]. Regardless of these effects on aquatic species, wetlands are still effective at removing solutes from water. Over the years, constructed wetlands have been used as a less-expensive alternative to store and treat stormwater and wastewater via adsorption and diffusion into pores [32–38]. Wetlands are also used to treat agricultural and industrial water by removing nutrients, trace metals and other substances [39, 40].

For this study, we focus on the transport of chloride within the Kampoosa Bog drainage basin, located in Stockbridge and Lee, Massachusetts. Kampoosa Bog is a calcareous lake-basin fen. Two major highways cross the watershed, Interstate-90 (Massachusetts Turnpike), and U.S. Route 7 (US-7) and are the major sources of road salt every winter. As of August 1995, Kampoosa Bog was designated as a Massachusetts Area of Critical Environmental Concern (ACEC) due to rising concerns about the effects of land development on the wetlands' water quality and ecology [41]. Since then, several studies have monitored water quality and its effect on plant diversity. [42] investigated the relationship between surface and groundwater

chemistry and the spatial distribution of *Phragmites australis* by installing piezometers and sampling species cover in the fen. In their 2004–2007 sampling period, Rhodes and Guswa [25, 43] studied the effects of hydrological events like precipitation and snowmelt on storage of sodium and chloride in the fen.

Vegetation studies done at Kampoosa show an increase in invasive and opportunistic species such as cattail and common reed that have colonized regions of the wetland and led to reductions in occurrences of some rare and more vulnerable plant species [25, 42, 44]. Vegetation restoration efforts by the Massachusetts Division of Fisheries and Wildlife Natural Heritage and Endangered Species Program and the Nature Conservancy in 2008 to treat the *Phragmites* were successful; however, cattails have since predominated the wetland in the treated areas thus still changing the plant community [44]. In 2017, Mays completed a plant community composition survey for Natural Heritage and Endangered Species Program, under the Massachusetts Division of Fisheries and Wildlife (NHESP; [44]) to document the evolving ecology. The toxicity of sodium and chloride to plants has been studied extensively [45, 46]. Salt can interact with other factors to influence plant growth. For example, (1) sodium and calcium interactions can lead to decreases in calcium uptake by the roots, and (2) salt can increase soil pH which can lower the availability of other soil nutrients such as phosphorus [46]. Additionally, an increase in sodium leads to decreases in biomass accumulation, inhibits plant growth and promotes crop senescence [47, 48].

These studies at Kampoosa Bog provided evidence for chloride contamination, a major concern for both ecologists and the Massachusetts Department of Transportation (MassDOT). As a result, the NHESP and MassDOT collaborated to fund a more detailed long-term hydrologic study aimed at monitoring the water quality at Kampoosa Bog. In addition to being an ACEC, Kampoosa Bog is located in a relatively small watershed making it an ideal study site. Moreover, the watershed allows for the establishment of multiple monitoring sites, providing a more detailed overview of the water quality. Road salt application data from the MassDOT enables the determination of chloride influx and net accumulation in the system.

From 2017, the Groundwater/Road Salt ISA Group from the University of Massachusetts—Amherst has been continuously monitoring surface and groundwater quality and streamflow using gauge stations and by sampling for groundwater chemistry with monthly grab samples. In 2021, Rhodes et al. [49] examined the data collected by the Groundwater/Road Salt ISA Group. Their analysis suggested that dissolved chloride concentrations in the fen groundwater appear to be statistically similar between the 2004–2007 [25, 43] and the 2017–2020 sampling periods, however, differences in sampling protocols brought this conclusion into question. Therefore, a different approach is needed to evaluate if dissolved salt concentrations are increasing at Kampoosa Bog.

In this study, we examine chloride storage and mass flux at Kampoosa Bog over a two- and half-year period (2018–2020). We investigate how modifications to salt application rates between 2012 and 2019 influenced the groundwater concentrations. Our primary hypothesis is that, over short periods, changes in road-salt application rates do not affect the groundwater concentrations in the watershed. We predict there is a delayed response time between changes in application rates and watershed chemistry concentrations and expect discharge to be a primary driver of fluctuations in chloride export. We also compare the behavior of the three subwatersheds in the drainage basin with regards to hydrology and sodium and chloride storage and fluxes. The function of wetlands in road-salt contaminated areas is assessed by comparing chloride retention rates in two subwatersheds. The first subwatershed includes the Kampoosa Bog fen region, which has higher wetland percent cover, and a second subcatchment (KB150) that has lower percent wetland cover. We predict that percent wetland cover is correlated with chloride retention efficiency; chloride retention is increased by greater wetland cover because

of lower flow through these landscape features. For our analysis, we use discharge, chloride mass flux, precipitation and road-salt application data collected between January 2018 and April 2020 to study annual, monthly, and daily patterns of chloride storage and release.

## Study area

The Kampoosa Bog drainage basin (Fig 1) is a 465 ha watershed located in western Massachusetts (42.293 N, 73.305W) that comprises ponds, a graminoid fen, shrub fens and red maple swamps [25, 26]. Springs at the base of the Rattlesnake Mountain supply water to Marsh Brook, which flows southeast under Rattlesnake Mountain Road and through a culvert under

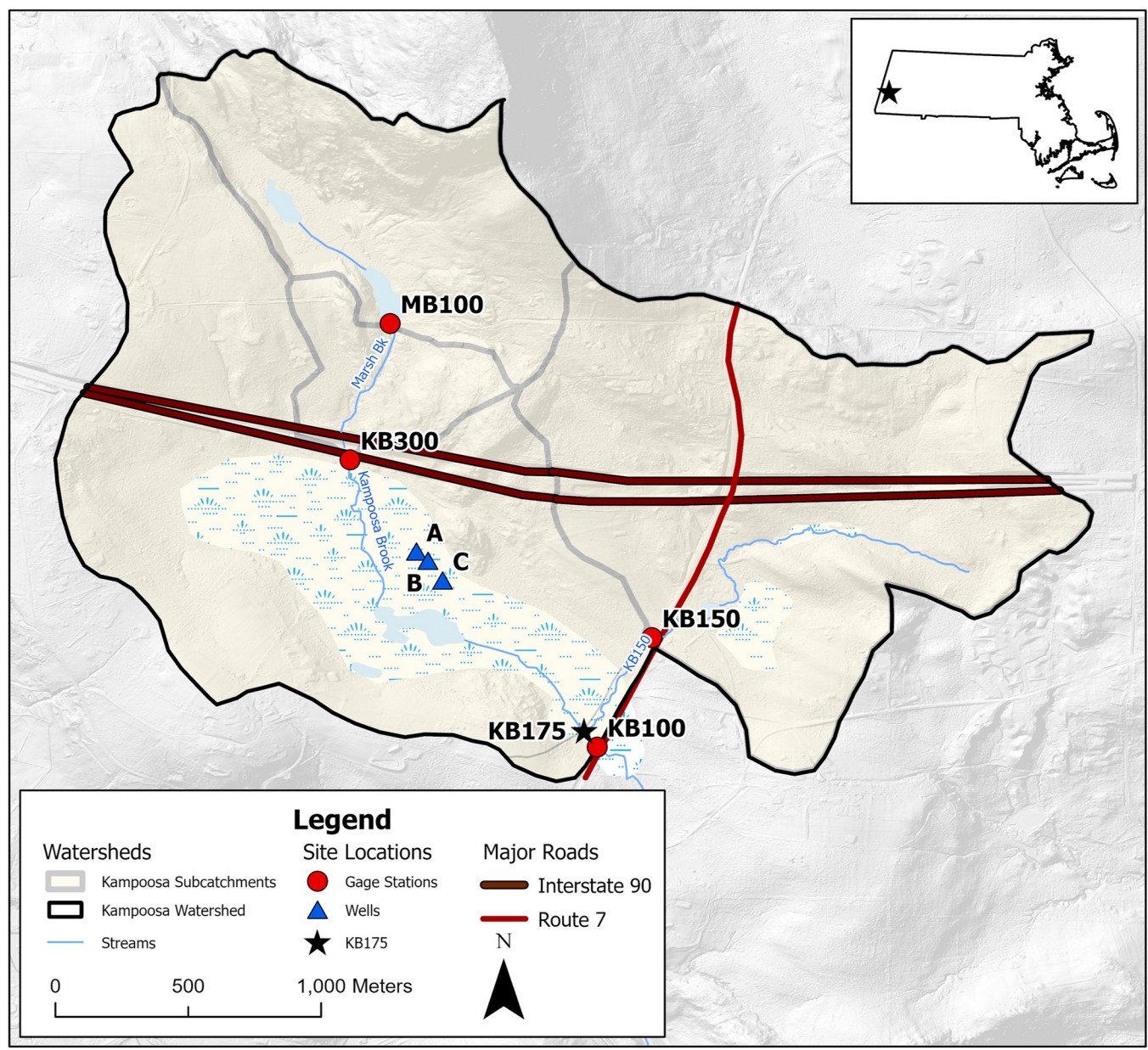

**Fig 1. Map showing Kampoosa Bog basin sub catchments and sampling sites.** The map was created using ESRI ArcPro software based on public domain aerial photography and ESRI digital elevation models.

the four-lane I-90 before entering the calcareous fen; Marsh Book is renamed Kampoosa Brook at the culvert (Fig 1). The fen region is dominated by groundwater inputs with some surface water inputs as well. East of this fen region, there is a second subcatchment and an unnamed stream which is located south of I-90. The stream flows southwest and parallel to the two-lane US-7 state highway before joining Kampoosa Brook upstream of the outlet culvert of the entire watershed at a point named KB100.

Four gauged subcatchments comprise the KB100 (465 ha) watershed which drains at the KB100 gauge, just upstream of a culvert underneath US-7. The smallest subcatchment is MB100 (81 ha) located in the northwestern region which is gauged in the culvert under Rattle-snake Mountain Road. MB100 serves as a reference watershed for chloride concentrations because it is upstream of the state and interstate highways. The KB300 watershed (106 ha), which includes the MB100 watershed, drains into the fen region and receives small amounts of chloride from I-90. Its gauge station is located at the culvert exit on the southern side of I-90 and on the northern boundary of the calcareous fen. To the east, the KB150 watershed (167 ha) receives salt from both I-90 and US-7. The gauge for this watershed is located 3m down-stream and just west of a culvert under US-7, east of the fen region. The tributary from this subwatershed joins Kampoosa Brook just upstream of the KB100 gauge. The KB150 watershed has a wetland percent cover of 8.2%.

The Kampoosa Bog fen is the largest calcareous lake-basin fen remaining in Massachusetts. It has important ecological significance as it provides habitat for at least 19 state listed rare and endangered species [25, 26, 41, 44]. There are over 60 plant species at Kampoosa Bog; and these include sedges (*Carex aquatilis*, *C. lasio-carpa Ehrh.*, *Cladium mariscoides* [Muhl.] Torr.) and low shrubs (*Betula pumila L.*, *Myrica gale*, *Potentilla fruticosa L.*, *Vaccinium macrocarpon*; [26, 44]).

Kampoosa Bog has been an area of interest in western Massachusetts over the past decades because of the need to conserve the diverse plant communities in the fen. Understanding the fate and transport of road salt applied to the roads in this watershed is a critical component of managing this wetland. Since 2017, collaborative efforts by the MassDOT and the Massachu-setts Department of Environmental Protection have led to the intensive studying of Kampoosa Bog with most of the focus being on water and soil quality studies. The two state agencies have shown great interest in understanding the impacts of water and soil contamination on species diversity by providing funding for the development of a water-quality monitoring study. How-ever, the water-quality data collected in this watershed during the monitoring period have not yet been analyzed to provide scientific insights on the impacts of road salt on water quality. The purpose of our work is to address this need by conducting mass-balance analyses of the chloride to improve understanding of the watershed's response to salting practices. Our study provides insight to the accumulation and flux of chloride in small watersheds and wetlands. Moreover, the role and function of wetlands in road-salt impacted watersheds is explored by quantifying chloride retention rates and their impacts on water quality. This research contrib-utes towards the understanding of wetland ecosystems by highlighting their significance in road-salt polluted watersheds. Specifically, we provide insights on what happens to the salt applied to the I-90 and US-7 highways.

## Materials and methods

### Ethics statement

The majority of the Kampoosa Bog wetland spans privately owned land. Access to field sites for this project was arranged with permission by the Massachusetts Department of Fisheries and Wildlife, Natural Heritage Endangered Species Program and by the Massachusetts Depart-ment of Transportation.

## Data collection and sources

**Hydrologic gauges.** To monitor water quality, the Groundwater/Road Salt ISA Group from the University of Massachusetts—Amherst designed a field study and deployed four gauges at MB100, KB300, KB150 and KB100. The MB100 site, located north of I-90, was chosen to provide an overview of the typical water quality in regions without major highways. Conversely, KB300 and KB150 were considered the primary sampling points for chloride applied to I-90 and US-7, respectively. Both KB300 and KB150 were chosen due to their proximity to these major highways and easier accessibility. The KB100 gauge station was strategically located downstream of the fen region along Kampoosa Brook where all the exported chloride could be measured.

At the four gauges, water level and temperature (˚C) plus specific conductance (μS/cm) were measured every fifteen minutes from Nov 2017 to Oct 2020 using a HOBO U20-001-04 water level data logger and a HOBO U24-001 conductivity data logger (Onset Computer Corporation; Bourne, MA), respectively. However, data from the MB100 gauge was excluded from the analysis due to conductivity probe malfunctions during this period. We focus on data collected from Jan 2018 to April 2020 due to (1) irregular data collected at KB150 caused by ice formation during Dec 2017 and (2) the presence of a beaver dam at the KB100 culvert from May 2020 to Oct 2020 which flooded the fen region and impacted the stage-discharge relationship. Other irregularities in the data caused by stream freeze-thaw cycles occurred over shorter time frames ranging from a few hours to less than 10 days and were addressed by interpolating between reliable data points. Interpolating between these data points may have failed to fully account for some of the complex variations in stage and discharge, which could have led to an oversimplified representation of the data and potentially obscured other short-term fluctuations, but we believe these to be minor given the short timeframe.

**Hydrogeochemical data—stream discharge and water chemistry.** Water quality data were provided by the Groundwater/Road Salt ISA Group. Additionally, streamflow (gpm) is periodically measured using a FH950 portable velocity meter (Hach Company; Loveland, CO). Surface water samples collected at the gauge stations between May 2017 and April 2019 were analyzed for major anions and cations. Across all four sites, chloride concentrations measured from grab samples were calibrated against specific conductance ($k$) readings and used to develop an empirical model of chloride concentration ($C_{Cl}$, mg/L; Eq 1).

$$C_{Cl}\left(\frac{mg}{L}\right) = 51.2k_{stream}^2\left(\frac{S}{cm}\right) + 190.0k_{stream}\left(\frac{S}{cm}\right) - 47.1 \tag{1}$$

**Hydrogeochemical data—grab sample groundwater and surface water chemistry.**
Monthly groundwater samples were collected inside the fen region from three sites by the Groundwater/Road Salt ISA Group (A, B, and C; Fig 1). At each site, 3 wells made from 2-inch PVC pipe with a 1-foot-long screen were installed with screen depths of 5 ft, 10 ft and 15 ft. Water samples were collected using a bailer and stored inside 120 ml HDPE bottles that were rinsed twice with the sample water. Surface water grab samples were also collected at each gauge (MB100, KB300, KB150 and KB100). These samples were then analyzed for Cl⁻ and other major ions by ion chromatography, in addition to pH, specific conductance and alkalinity.

**Meteorological conditions.** Stockbridge and Lee are characterized by warm, wet summers (June-Aug) and snowy winters (Dec—Feb). Between 2018 and 2020, average monthly temperatures throughout the year range between -9˚C in January and 27˚C in July (Fig 2). Annual rainfall is approximately 1300 mm/year while monthly totals range from 30mm (Jan 2020) to 226 mm (Aug 2018). Long-term data indicate that snow depths of 965 mm -1146 mm are received each year in the region [50].

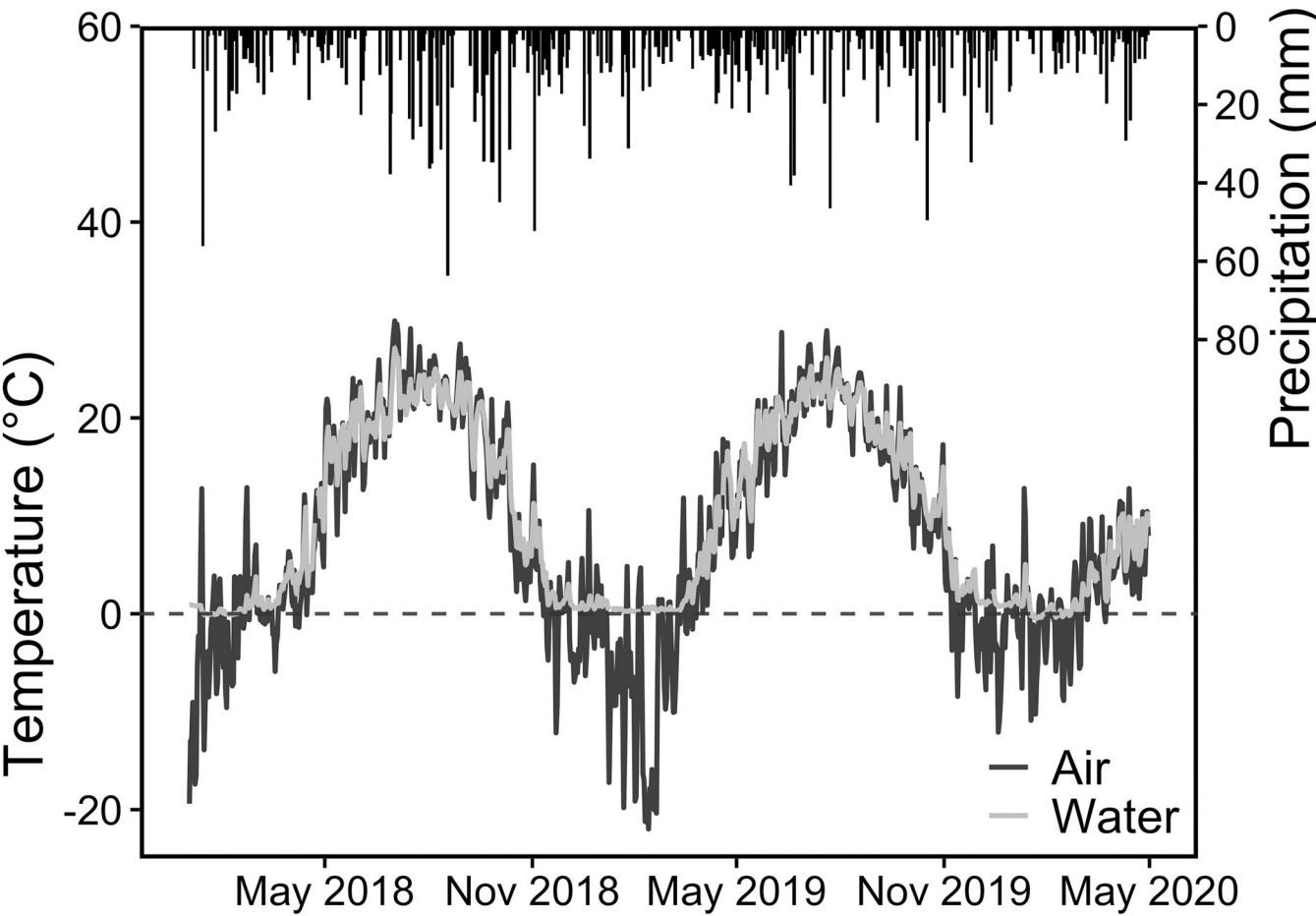

**Fig 2. Daily precipitation and mean daily air and water temperatures at Kampoosa Bog.** Water temperature from three gauges (KB-100, KB-150 and KB-300) is averaged and air temperature is used from the KB-100 gauge.

Precipitation (mm) data are from the MA-BE-3 weather station located in Stockbridge, MA [51]. Potential Evapotranspiration (PET, mm) is estimated by averaging monthly mean PET values for Albany, NY and Hartford, CT, calculated using a modified version of the British Meteorological Office Rainfall and Evaporation Calculation System (MORECS) model on data collected from 1981–2010 *(Northeast Regional Climate Center)*. Monthly average PET ranges from 9.0 mm in the cooler Dec–Feb months to 113.7 mm in warmer June–Aug months. Air temperature (˚C) is measured using a HOBO U20-001-04 data logger at each gauge station.

**Salt application data.** Road-salt application data from December 2012 to April 2020 were provided by the Massachusetts Department of Transportation (MassDOT). The MassDOT maintains the state highways: they record the amount of salt applied to the four-lane I-90 and two-lane US-7 in western Massachusetts for every winter storm. They provided salt application data for 20.8 miles and 19.27 miles of I-90 and US-7, respectively, near the study site. Chloride applied to each subwatershed is estimated by scaling the MassDOT data to the length of each highway within each subwatershed (Table 1). MassDOT notes that this may result in an over-estimation of salt applied within the Kampoosa watershed because other parts of the documented miles of I-90 are at higher elevations that may receive a greater proportion of the road salt. Additional uncertainty to these data could also be introduced by the uncertainty associated with estimating the road length in each subwatershed that is contributing salt.

**Table 1. Kampoosa Bog subwatershed areas, total length of state highways (I-90 and US-7) lanes and percent wetland cover.**

| Watershed Name | Area (ha) | Total Length of Highway Lanes (m) | Lane Density (m/ha) | Percent Wetland Cover (%) |
|---|---|---|---|---|
| Wetland Region | 294 | 5429 | 18.4 | 23 |
| KB300 | 96 | 1112 | 11.6 | - |
| KB150 | 166 | 8256 | 49.7 | 8 |
| KB100 | 465 | 14797 | 30.5 | 17 |

## Study design

**Chloride mass-balance and steady-state concentrations.** We characterize chloride storage ($Cl_{watershed}$) in the KB150, KB300 and KB100 watersheds by using a watershed mass-balance. The inputs are chloride applied to lanes of I-90 and US-7 that are in the watershed $Cl_{applied(I-90+US-7)}$ while the output ($M_{watershed}$) is the chloride mass flux measured at the watershed gauge. The MB100 and KB300 watersheds, located north of I-90, drain into the fen region in the KB100 watershed and are not major sources of sodium and chloride. Streamflow grab sample data from the MB100 (reference watershed) and KB300 gauges show low levels of chloride (Fig 3). Low chloride concentrations at KB300 indicate the presence of other pathways for chloride applied to I-90 to enter the fen and migrate to the outlet.

A Seasonal Mann-Kendall test is used to identify any trends in the monthly chloride budgets, where positive trends indicate chloride accumulation while negative trends indicate chloride dilution [52]. The cumulative monthly chloride storage is calculated using (Eq 2):

$$Cl_{watershed}^{j+1} = Cl_{watershed}^{j} + Cl_{applied(I-90+US-7)}^{j} - M_{watershed}^{j} \qquad (2)$$

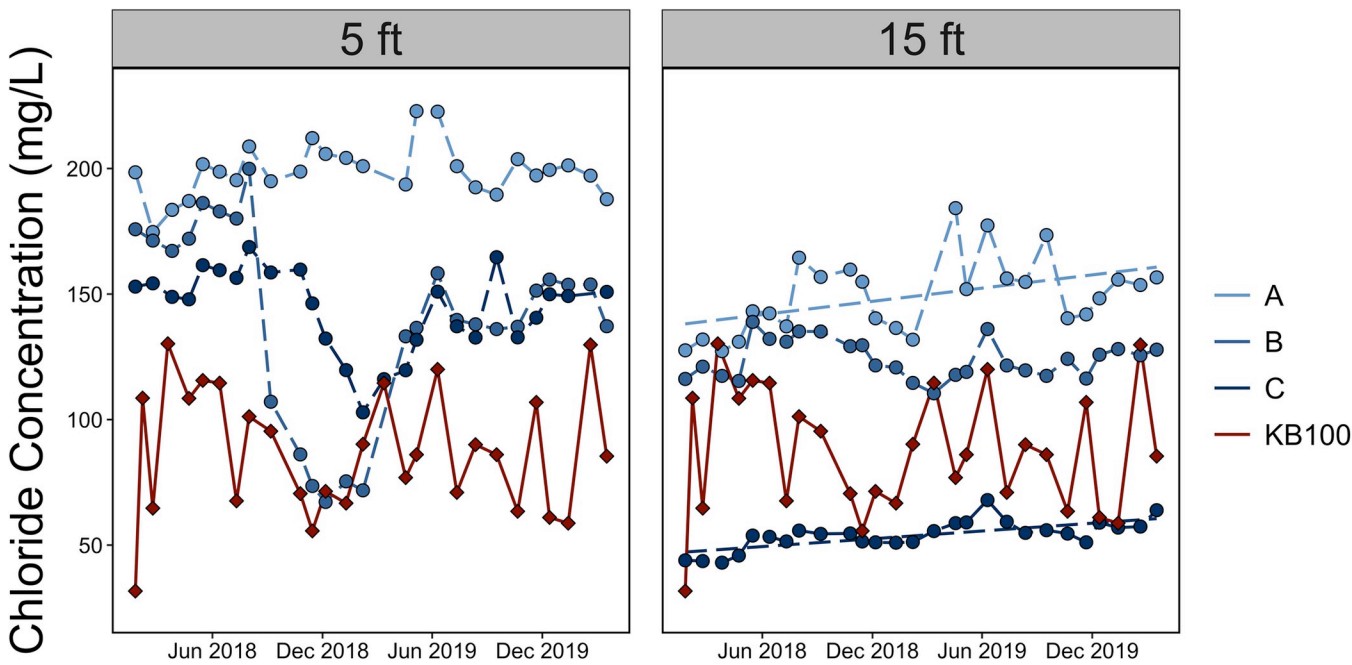

**Fig 3. Monthly groundwater (A, B and C) and surface water (KB100) chloride concentrations measured in the fen region of the Kampoosa Bog watershed.** Each subplot represents the depth (5 ft and 15 ft) at which the sample was collected, and color is used to represent the site (well A, B and C and KB100).

where ($Cl_{watershed}$) represents accumulated chloride at the start of any given month ($j$) or the subsequent month ($j+1$). $M^j_{watershed}$ is the mass of chloride that fluxes from the watershed while $Cl_{applied(I-90+US-7)}$ is the amount of chloride applied to lengths of US-7 and I-90 that are in the watershed. Due to the lack of historic data, the model was simplified by arbitrarily setting the initial $Cl^j_{watershed}$ in the watersheds to zero. Thus, calculated values of chloride storage are all relative to January 2018.

In addition to the mass-balance, we estimated a long-term steady-state chloride concentration for KB150 and KB100 as the total chloride applied between 2012 and 2017 normalized by the total streamflow over the same period. Since streamflow measurements were not available for this period, we used the water balance (Eq 3) to estimate annual flows at each gauge and subsequently the long-term steady-state chloride concentrations. We then compared these steady-state concentrations with those for the 2018–2019 period.

## Water balance

To quantify potential losses of surface water, and consequently chloride, to the deep subsurface, we calculated the change in water storage on the monthly timescale for the KB100 watershed using (Eq 3):

$$\frac{dS}{dt} = (I - ET)*Area - Q_{out} \tag{3}$$

Where $\frac{dS}{dt}$ is the change in water in storage, $I$ is precipitation, $ET$ is evapotranspiration, $Q_{out}$ the streamflow out of the watershed, and $Area$ the area of the watershed. An apparent net gain of water over time could indicate an unaccounted for loss of water to the subsurface. A closed water balance indicates that we have appropriately accounted for inflows and outflows.

## Role of wetlands

To assess the role of wetlands in road-salt contaminated watersheds, we compared the KB150 subwatershed (8% percent wetland cover) to the fen region in the KB100 watershed (west of the KB150 watershed) which has a higher wetland percent cover (23%, Fig 1). Since the fen region is part of the KB100 watershed, we first established a theoretical site (KB175) located upstream of the KB100 gauge station which isolates the chloride changes in the region. Deploying an actual gauge station at this location during the sampling period was infeasible due to its physical inaccessibility, so we calculated the chloride loading values at KB175 by subtracting the values recorded at KB150 from those recorded at KB100. We then calculated each watershed's annual retention capacity using (Eq 4):

$$Annual\ Retention\ Capacity\ (\%) = \left(\frac{Input - Export}{Input}\right)*100 \tag{4}$$

where the $Input$ is the total mass of chloride applied to the roads in the watershed and $Export$ is the total mass of chloride that is carried out of the watershed by discharge as determined at KB175 and KB150.

## Results

### Groundwater-surface water relationship in the fen region

Groundwater concentrations in the fen region exhibited distinct spatial variations where Cl concentrations were higher closer to I-90 and decreased as distance from the road increased, consistent with prior studies at Kampoosa Bog [25, 26]. This trend is evident in Fig 4 where

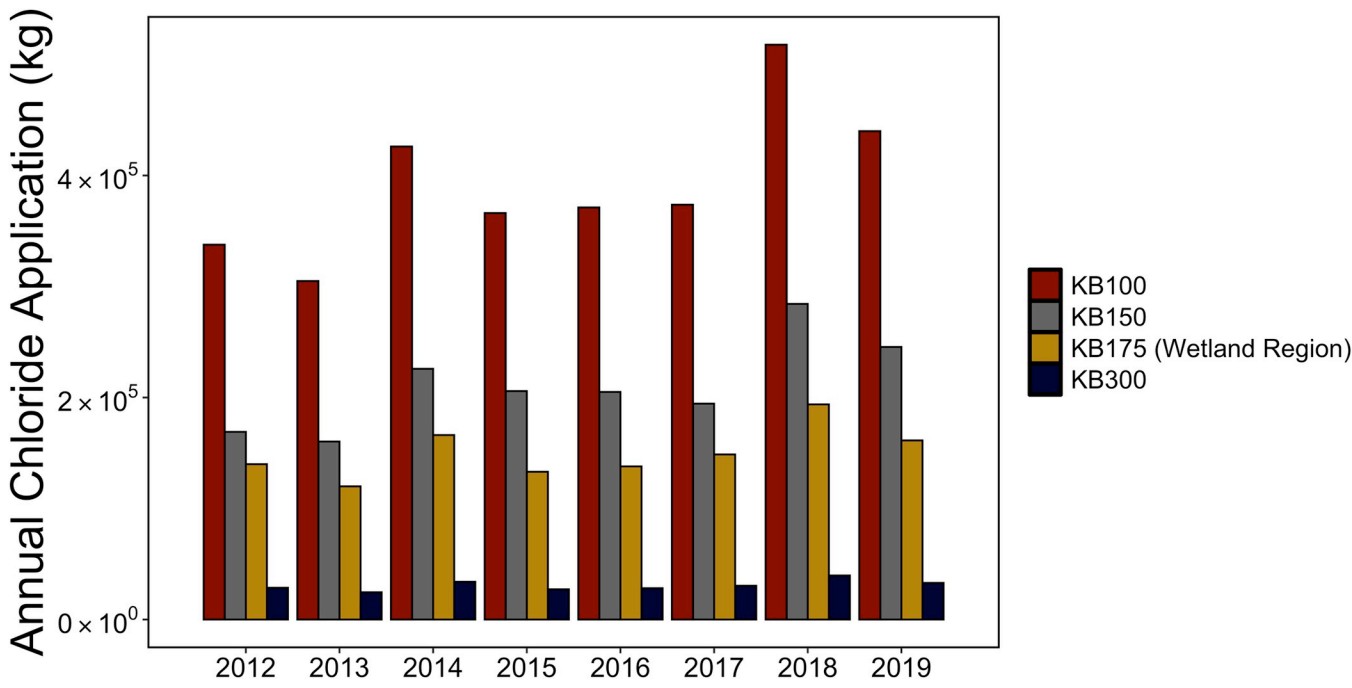

**Fig 4. Annual chloride application rates in the KB100, KB150, KB300 subwatersheds and the wetland region.**

the highest concentrations were recorded at well A and the lowest at well C. Additionally, the concentrations varied with depth and were generally greater in the shallow wells (5 ft) for all the three wells (A, B and C). Chloride concentrations ranged from 127 mg/L (15 ft) - 222 mg/L (5 ft) at A, 67 mg/L (15 ft) - 200 mg/L (5 ft) at B and 43 mg/L (15 ft) - 198 mg/L (5 ft) at C (Fig 3). Between Aug 2018 and Feb 2019, there was a prolonged dilution event in the shallow wells (5 ft) at the B and C sites which led to decreases in concentrations from 200 mg/L to 70 mg/L at B and 150 mg/L to 100 mg/L at C. These dilutions coincide with times of significant precipitation and discharge at KB100 (S5 Fig).

We observed changes in chloride levels at well A (15 ft depth) between Feb 2019 and April 2019 where concentrations increased by ~50 mg/L from an initial value of 131 mg/L (Fig 3). Well C (15 ft depth), also had an increase in chloride concentrations over the Feb 2019 –April 2019 period. Between Jan 2018 and April 2020, chloride concentrations increased by 23% and 45% at the 15 ft depth wells at locations A and C, respectively. In contrast, concentrations at the 5ft wells exhibited both increasing (A) and decreasing trends (B and C). From 2018–2020, mean groundwater chloride concentrations at each well, calculated as the average of the concentrations at the 5 ft, 10 ft and 15 ft depths, were 175 mg/L, 125 mg/L and 99 mg/L for wells A, B, and C, respectively. Rhodes and Guswa [25] report average concentrations for those same wells from 2002–2007 of 273 ± 61 mg/L, 184 ± 41 mg/L and 171 ± 30 mg/L. Streamflow at KB100 tended to be less saline than the groundwater collected from the wells as indicated by the lower chloride concentrations (30 mg/L– 130 mg/L; Fig 3). However, deep groundwater (15 ft) concentrations at well C were lower, and the maximum difference between the groundwater and surface water (KB100) concentrations was ~100 mg/L.

### Watershed responses to salt application practices

Road-salt application rates along I-90 and Rt.7 varied over the study period. Normalized annual chloride application rates for 2018 and 2019 were approximately 35 kg/lane-meter and

**Table 2. Summary of annual discharge and chloride mass-balances for the Kampoosa subwatersheds for 2018 and 2019.** The road lengths in each subwatershed are used to normalize the chloride inputs. Precipitation in 2018 is 1552 mm and 1332 mm in 2019 while mean annual PET is ~ 640 mm.

| Year | Watershed Name | Discharge (mm/year) | Chloride Input (kg) | Chloride Outflux (kg) | Chloride Accumulation (kg) | Retention Efficiency (%) |
|------|----------------|---------------------|---------------------|-----------------------|----------------------------|--------------------------|
|      | Wetland Region | 696 | 194000 | 144000 | 50000 | 26 |
|      | KB150 | 872 | 284000 | 203000 | 82000 | 29 |
| 2018 | KB300 | 877 | 40000 | 26000 | 14000 | 35 |
|      | KB100 | 934 | 518000 | 372000 | 146000 | 28 |
|      | Wetland Region | 411 | 161000 | 63000 | 98000 | 61 |
|      | KB150 | 662 | 246000 | 152000 | 93000 | 38 |
| 2019 | KB300 | 548 | 33000 | 15000 | 18000 | 54 |
|      | KB100 | 611 | 440000 | 230000 | 210000 | 48 |

30 kg/lane-meter. Historic road salt application data indicated that salt application rates were slightly higher in 2018 and 2019 compared to 2012–2017 (Fig 4 and S1 Table). In 2018, about 40000 kg, 284000 kg and 518000 kg of chloride were applied to the roads in the KB300, KB150 and KB100 watersheds, respectively. Although there were reductions in the chloride application rates in 2019, the rates were still relatively high. The 2019 application rates were the second highest (440000 kg) between 2012–2019 followed by 2014 where 430000 kg of chloride were applied to the roads in the KB100 watershed. In the KB100 watershed, the 2018 and 2019 salt applications were 43% and 21% greater than the average annual applications from 2012–2017.

Table 2 represents the annual chloride application, outflux, and retention for each subwatershed in 2018 and 2019. Chloride exports exhibited a comparable trend to the application rates with higher levels being recorded in 2018. In contrast, chloride accumulation rates were lower in 2018 and higher in 2019 (Table 2). Across the four subwatersheds, chloride retention efficiencies ranged from 26% in 2018 to 61% in 2019. Although the KB300 subwatershed has no known wetlands, we estimated relatively high retention rates during 2018 and 2019. This could be due to the uncertainty introduced during the watershed delineation resulting in the overestimation of the road length contributing road salt to the KB300 subwatershed.

To complement the annual results, we analyzed the data at a monthly scale from Jan 2018 – April 2020 (Fig 5). The results from the chloride mass-balance are consistent with the idea that the KB300 watershed is not a major source of contamination to the KB100 watershed (Fig 5 and S9 Fig). We observed decreases in chloride application rates in 2020 (Jan to April) where approximately 8000 kg, 62000 kg and 106000kg were applied to the KB300, KB150 and KB100 watersheds, respectively. Monthly chloride application rates were generally higher in February and March and lower in April at the end of each snow and ice season. Monthly chloride application rates tended to be less than 7000 kg/month, 60000 kg/month, and 100000 kg/month in the KB300, KB150 and KB100 watersheds, respectively. However, over 11000 kg, 70000 kg and 130000 kg were applied to these watersheds in Dec 2019. Overall, there is evidence for net chloride accumulation in all the subwatersheds from Jan 2018 –April 2020.

While chloride applications were limited to the snow and ice months (late Oct–May), chloride exports occurred throughout the year and were more pronounced during the Oct–April months (Fig 5). Chloride outflux rates were typically less than 4000 kg/month, 28000 kg/month and 50000 kg/month for the KB300, KB150 and KB100 watersheds respectively. Outflux displayed a distinct cyclical pattern where the lowest values were recorded during the dry months (June-Aug). These outflux rates then steadily increased between Sept and Nov before reaching peak values in Dec through Feb and then decreasing from March to April. At the end of April 2020, we estimated that 36000 kg, 188000 kg and 339000 kg of Cl had accumulated in

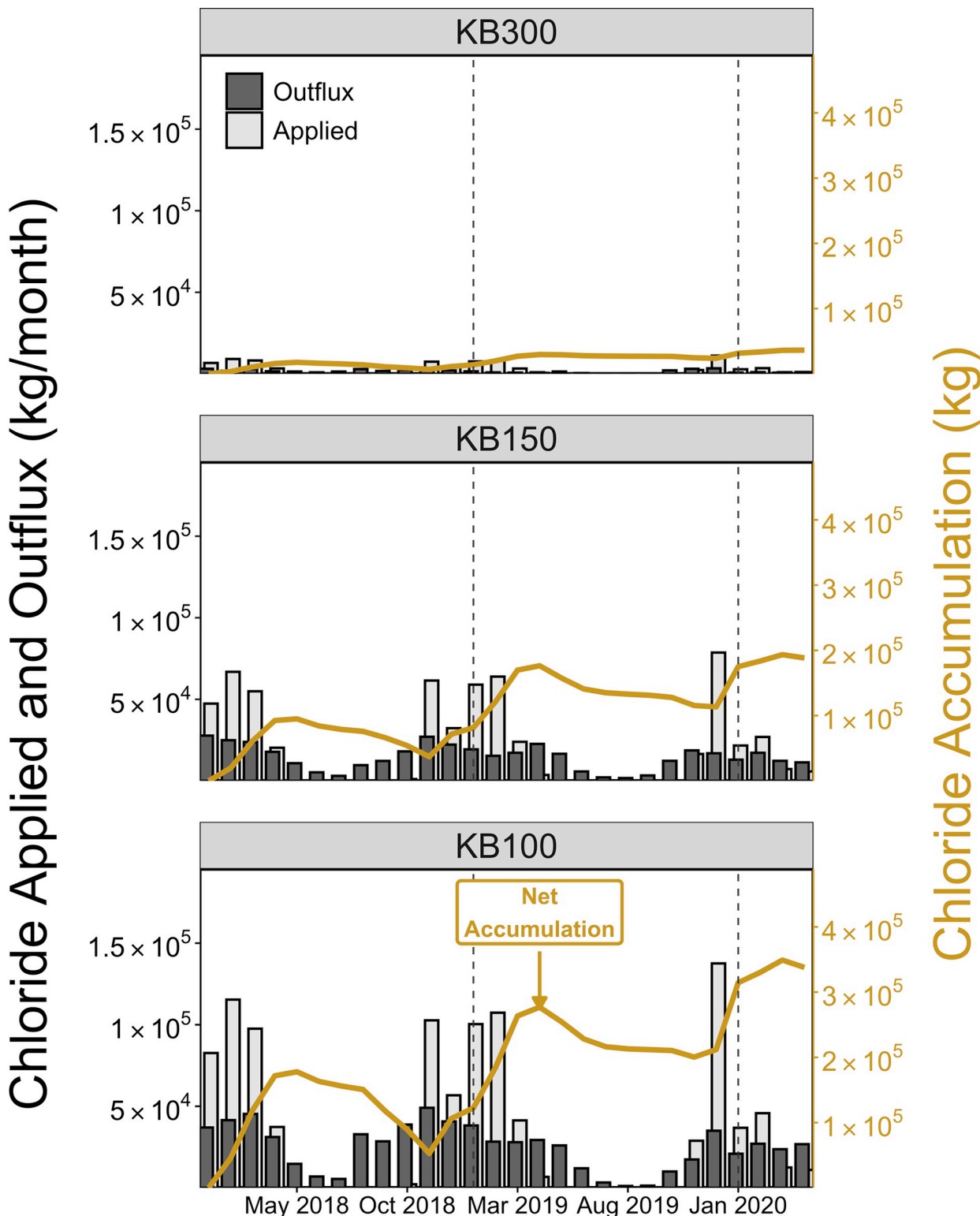

**Fig 5. Monthly chloride mass-balance for the KB300 (top row), KB150 (middle row) and KB100 (bottom row) subwatersheds between Jan 2018 and April 2020.** The light gray bars represent the total monthly chloride applied to the roads in each watershed and the dark gray bars the monthly chloride outflux. The secondary axis on the right and solid line represents the accumulation of chloride in each watershed since Jan 2018.

the KB300, KB150 and KB100 watersheds since the beginning of the study period representing 49%, 35%, and 35% of the applied chloride, respectively. The mass of chloride in storage decreased during the non-salting months, indicating chloride export. The highest chloride net export (negative slope) occurred in 2018 (Sep–Nov) and coincided with large outflux events. We were able to detect decreasing trends ($\tau$ = -0.005, $\tau$ = -0.074, $\tau$ = -0.027) in monthly Cl accumulation values in the KB300, KB150 and KB100 watersheds; however, the Seasonal Mann Kendall tests showed that these trends were not statistically significant ($p > 0.05$). This indicated that there were no monotonic trends in the monthly chloride accumulation data.

Despite the evidence for chloride accumulation, the water balance indicates that there is no significant net accumulation or loss of water in the KB100 subwatershed relative to the precipitation and evapotranspiration (S8 Fig). The water balance is closed as indicated by the similar water storage values at the beginning and end of the study period. The net accumulation of water from Jan 2018 –April 2020 was 128000 m$^3$, which is equal to only 0.9% of the precipitation volume over the same period.

## Watershed responses to streamflow variations

Flow-duration curves of normalized discharge from KB300, KB150 and KB100 are similar (Fig 6A). All subwatersheds are dominated by low streamflow with flows equal to or exceeding 4 mm/day occurring only 10% of the time. However, the low flows in the KB100 watershed tend to be lower than those of the KB150 and KB300 subwatersheds. Daily median flows are 1.5 mm/day at KB150, and 1.6 mm/day at KB100 and KB300.

There is a strong linear relationship between daily discharge and chloride mass flux in the KB100 and KB150 watersheds (S1 Fig), and in both watersheds more than 70% of the variation in chloride flux was explained by discharge. The data were split into four periods (SON: Sept-Oct-Nov, MAM: March-April-May, JJA: June-July-Aug, DJF: Dec-Jan-Feb) to analyze seasonal trends. Like the monthly trends, flux was higher from Sept–Feb, with a few large outflux events occurring between June and Aug (S1 Fig). The KB300 watershed did not display any trends between the discharge and mass flux: low flow events led to both high and low outfluxes particularly between Sept and Feb (S1 Fig).

Using chloride flux concentrations which quantify the total mass of chloride per total volume of water over a period, we observed the highest flux concentrations were recorded at the KB150 gauge (Fig 6B). Overall, median concentrations of 23 mg/L, 160 mg/L and 96 mg/L were recorded at KB300, KB150 and KB100, respectively. Despite these variations, chloride

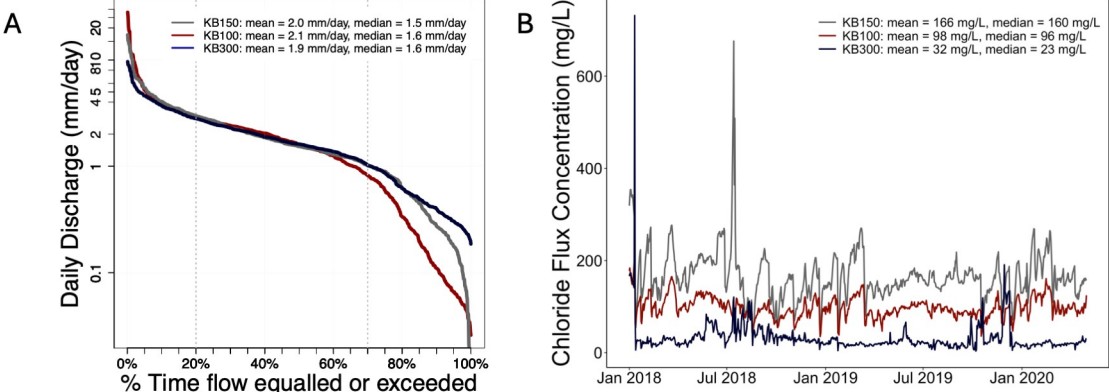

**Fig 6.** A: Flow duration curves for the KB300, KB150 and KB100 watersheds normalized by the watershed area. B: Daily chloride flux concentrations measured at the gauge stations in each watershed.

flux concentrations at KB100 and KB150 followed a similar temporal pattern as shown by the almost synchronized increases and decreases in concentrations, driven by hydrologic events. Relatively lower chloride outfluxes (S1 Fig) and high discharge at KB300 lead to lower chloride flux concentrations.

Chloride mass fluxes normalized by the lane meters of salted roads further showed that the KB150 and KB100 watersheds respond similarly to road-salt (S3 and S7 Figs). Mean (0.058 kg/day/m) and median (~0.052 kg/day/m) values for these two watersheds were similar. Moreover, both watersheds followed a seasonal pattern characterized by higher mass fluxes in the Dec—April months (Fig 6B).

### Role of wetlands in road-salt contaminated areas

To understand the importance of wetlands in salt contaminated watersheds we quantified and compared chloride retention rates for two subwatersheds with varying wetland percent cover. Chloride retention rates in the KB150 watershed and wetland region, located west of KB150, differed between 2018 and 2019 (Table 2). The retention efficiencies were lower in 2018 in both regions. In 2018, retention efficiencies were comparable for the KB150 (29%) watershed and wetland region (26%). However, in 2019, the wetland region had a higher retention efficiency (61%) compared to the KB150 watershed (38%).

### Long-term and short-term steady-state chloride concentrations

During the Jan 2018 –April 2020 period, the water-balance models for the KB150 and KB100 (S8 Fig) subwatersheds indicated there was an insignificant net accumulation of water. As a result, the annual streamflow was estimated as the difference between the precipitation and evapotranspiration. The performances of these water balance models were high across the two subwatersheds (NSE > 0.93, KGE > 0.86, $r^2$ > 0.97). Using this water balance model approach, we simulated annual flows and long-term steady-state chloride concentrations for the 2012–2019 period. The observed average annual chloride loading during this period was 211000 kg and 392000 kg for the KB150 and KB100 subwatersheds, respectively. The KB150 and KB100 subwatersheds had steady-state concentrations of 250 mg/L and 166 mg/L, respectively. Using the observed streamflow and chloride application rates from 2018 to 2019 (short-term), we calculated slightly lower steady-state concentrations of 207 mg/L and 133 mg/L for the K150 and KB100 subwatersheds, respectively. Even though salt application rates were higher than average in 2018 and 2019, the wetness of those years (P = 1552 mm and 1332 mm, respectively, versus 1151 mm average for 2012–2019; S1 Table) leads to a lower steady-state chloride concentration for 2018–19. The observed net accumulation in 2018 and 2019 is due to the delay in the transport of that salt through the watersheds; i.e., greater than average amounts of salt were applied, and the high precipitation and streamflow during those years was not able to flush it through the system.

## Discussion

### Effects of changes in salt application rates on groundwater chloride concentrations

The Kampoosa Bog drainage basin is impacted by road-salt contamination. Our study shows groundwater and streamflow concentrations in this region exceed the naturally occurring and background levels (Fig 3). Despite reductions in chloride application rates by ~15% in 2019 (between 2018 and 2019) in the fen region, the decreases in chloride concentrations at the 5 ft wells over this timeframe were not statistically significant. At well B and C, we observed

statistically insignificant (p > 0.05) concentration reductions which were due to prolonged dilution caused by high precipitation events in Aug 2018. The water table in peatlands, like the Kampoosa fen region, is close to the surface which allows precipitation and evaporation to dilute and increase the concentrations, respectively [53, 54].

Shallow groundwater concentrations measured at Well A (5 ft depth) remained similar even though chloride concentrations at the surface decreased (S4 Fig). We speculate that differences in peat permeability caused by varying vegetation type in the acrotelm (upper-peat layer) could have resulted in varying spatial trends in the hydraulic conductivity of peat. The latest in-depth vegetation identification study at Kampoosa Bog documents the presence of different groups of plant species close to the wells [44]. Well A appears to be in a region dominated by bog birch (*Betula pumila)* shrub types which are known to have shallow roots while wells B and C are in a sedge (*Carex lasiocarpa-Carex aquatilis)* rich region. Sedges generally have longer roots which improve water infiltration rates. As a result, we hypothesize that the presence of shallow rooted plants at well A is indicative of an impediment to infiltration which then prevents the dilution of chloride solutes at greater depths. More detail about the current plant density and coverage in the fen could provide better insights on variations in the permeability of the peat.

We also observed significant increases in concentrations at the 15 ft depth wells between Jan 2018 to April 2019 (A and C; Fig 3). Increases in groundwater concentration are driven by chloride accumulation. Although there were reductions in road salt applications in 2019, the salting rates during the 2018–2019 period were much higher (32% increase) than those of the 2012–2017 period (Fig 4). Our analysis shows that changes in salt applications and watershed responses are out of phase. The presence of upward trends in chloride concentrations after reductions in salt application rates in 2019 indicate a lag between changes in salting practices and dissolved salt concentrations in the fen region. Delayed response times have been attributed to several attenuation processes and a decadal response time has also been proposed for other watersheds [20, 53, 55–58]. Similar decadal response times to change in salt applications on roads could be occurring at Kampoosa Bog.

When chloride is transported in and out of the fen region, two primary mechanisms can be used. The first is through surface water pathways which allow the contaminants to immediately exit the fen. The second is through slower groundwater movement; chloride is transported to the 5 ft and 10 ft depth wells primarily through groundwater [23]. At these wells, we expect advection, dispersion, and diffusion to be the primary physical processes responsible for the migration of chloride [34, 53, 56]. Since the fen is dominated by peat which is a dual porosity medium with active (allows easy transport of water and solutes) and inactive (negligible flow velocity) regions, diffusion allows exchange of solutes to occur between the two regions [34, 56, 59, 60]. The occurrence of diffusion decreases the solute movement which results in delayed reflections of reductions in salt application [56]. Despite the absence of hydraulic conductivity, head gradient and groundwater velocity measurements which are used to determine the groundwater and plume velocities, other studies have shown that peat can retard chloride due to the dual-porosity structure of its matrix and low hydraulic conductivities [56]. Additionally, through diffusion, the closed pores in the inactive region in the peat also act as a sink for chloride during the salting period, but over time become a chloride source when concentrations in the active region decrease [34]. We propose these mechanisms could be responsible for the consistently high chloride concentrations at the 5ft and 15ft depth wells over time.

From the mass-balance calculations, we observed a net accumulation of chloride in all the watersheds which was also evident in the deeper groundwater concentrations (15 ft depth). Increases in salting rates for both 2018 and 2019 led to increases in chloride accumulation and

groundwater chloride concentrations. Although groundwater was sampled only at three sites located in the central part of the fen region, we hypothesize that deicing salts applied to I-90 are also accumulating closer to the road leading to higher concentrations in this area. In a field-scale experiment involving NaCl transport in a bog, most of the contaminant plume remained near the spill point despite the plume advancing in the longitudinal direction near or at the top of the water table [34, 61]. Howard and Haynes [18] and Tiwari and Rachlin [62] have also suggested that groundwater can act as a sink for chloride. Additionally, Richburg et al. [26] reported higher chloride concentrations in the Kampoosa fen region closer to I-90, which ranged from 210 mg/L to 275 mg/L and decreased to 60 mg/L approximately 650 m from I-90.

The water balance also provides more evidence for chloride accumulation in the watershed. The water inputs and outputs are nearly balanced, suggesting the absence of potential unidentified outlets of water and dissolved chloride such as deep groundwater. Sampling of the groundwater closer to I-90 at varying depths can help determine the chloride concentrations and dimensions of the plume.

## Watershed responses to changes in chloride application and streamflow

Although the subwatersheds (KB300, KB150 and KB100) that make up the Kampoosa Bog drainage basin are of varying sizes, their responses to changes in salting practices and streamflow patterns are similar (Fig 6B, Table 1). A 15% reduction in chloride application in the three subwatersheds was observed between 2018 and 2019 (Fig 5). However, there were increases in the annual accumulation rates during that period due to lower precipitation and streamflow in 2019 compared to 2018 (Fig 2 and S5 Fig). Particularly, August 2018 was the 11[th] wettest August in Massachusetts over the 1895–2018 timeframe, and this resulted in high streamflow during that year [63]. At the monthly scale, high precipitation during the SON, DJF and early MAM period led to increases in streamflow and greater chloride export (S5 Fig).

We also observed the effects of groundwater-surface water interactions: streamflow observations from Kampoosa Brook which flows through the fen region, indicate that during the late MAM to JJA period when evapotranspiration is high, stream discharge is sometimes higher upstream (KB300) than downstream at the watershed exit point (KB100). This indicates that the stream is losing water to the fen and evapotranspiration during these periods (S2 Fig).

Streamflow also plays a critical role in exporting chloride from the watersheds [20, 64]. We observed a strong linear relationship between streamflow and chloride mass flux at two of the watersheds (KB150 and KB100), while the other (KB300) did not have any obvious trends (S1 Fig), and events where mass fluxes >2000 kg/day occurred were more common in the SON and DJF periods. While increases in chloride surface-water concentrations have been observed in other rural watersheds [20], we observed weak decreasing trends in chloride concentrations in the KB150 and KB100 watersheds (Fig 6B) which matched the 5 ft depth groundwater chemistry trends. The concentrations also followed a distinct seasonal pattern and were higher during low flows (S7 Fig). Chloride concentrations >100 mg/L occurred during all the seasons, but concentrations were generally higher in the DJF months (S6 Fig). High concentrations during non-salting periods are attributed to groundwater-surface water interactions where saline groundwater recharges the streams as baseflow [8, 65, 66]. Precipitation during this period dilutes concentrations and increases streamflow resulting in high mass fluxes (Fig 6B, S1 and S7 Figs). In the DJF period, high mass fluxes are due to high streamflow and road-salt applications. Although DJF months are dominated by snowfall, snowmelt events can increase discharge resulting in high chloride mass fluxes and some dilution of the concentrations.

## Role of wetlands

Detailed mechanisms for fate and transport of chloride in peatlands are discussed in 5.1. Contrary to our hypothesis which proposed a correlation between wetland percent cover and chloride retention, our study shows that the relationship is more complex (Table 2). Despite the KB150 subwatershed having a smaller wetland percent cover, chloride retention efficiencies in the two wetland regions were similar in the wet year (2018) while they were different in the dry year (2019). Chloride retention in the wetland region was 26% in 2018 and increased to 61% in 2019 while chloride retentions of 29% and 38% were observed in the KB150 watershed in those years.

Watershed responses were similar in 2018 despite the differences in watershed area and percent wetland cover in the two watersheds. This can be attributed to the similar high flow patterns observed in both watersheds (Fig 6A). However, there were differences in the low flow patterns recorded in 2019. The flow-duration curve shows that low flows in the KB100 watershed, which includes the wetland region, tend to be lower than those in the KB150 subwatershed. This pattern may be due to the topographic properties of the watersheds which influence the amount of water accumulating and flowing out of the watersheds. Watersheds with larger areas and gentle slopes tend to promote the accumulation of water while smaller watersheds with steep slopes facilitate faster surface runoff. At Kampoosa Bog, the 23% difference in retention efficiencies observed in 2019 can be attributed to (1) differences in wetland features, (2) differences in slope and (3) dry meteorological conditions. In the KB100 subwatershed, which includes the wetland region, there is one large continuous wetland feature through which Kampoosa Brook flows, whereas in the KB150 region, there are several smaller interconnected wetlands that are primarily riparian. We propose that the larger wetland in the KB100 watershed may play a significant role in slowing down the movement of water. Gentle gradients in the bigger wetland region promote the accumulation of water. On the other hand, the steeper gradient of the KB150 subwatershed also aids discharge and, in turn, the outflux of chloride leading to relatively low retention rates. Lower precipitation in 2019 resulted in lower outflows from this wetland region. Evidence for lower discharge is also shown by the increase in the number of days in which discharge at KB100 was less than that at KB150, increasing from two days in 2018 to 65 days in 2019. Since KB150 is a tributary to KB100, we would expect this mechanism to occur only when the wetland region functions as a water sink.

In addition to topography and meteorological conditions there is evidence that wetland location within a watershed also impacts streamflow routing [67]. Wetlands located further downstream have a greater potential to reduce flooding by lowering peak flows compared to headwater wetlands [67]. Headwater wetlands respond rapidly to precipitation events, including normal events, which decreases their buffering capacity [68] leading to generally higher flows. On the other hand, downstream wetlands tend to maintain a large buffering capacity. The wetlands in the KB150 subwatershed are located close to the headwaters while those in the fen region are further downstream in a larger basin. We observed that low flows in the KB150 subwatershed were generally higher compared to those in the wetland region (Fig 6A). The KB150 wetland provides continuous streamflow, even during drier years and consequently results in higher chloride outfluxes. On the other hand, the large flat expanse of lowland found in the KB100 subwatershed creates an opportunity for significant loss of water to evapotranspiration and concomitant reductions in streamflow and chloride export.

## Steady-state conditions and implications for ecology

For any watershed, steady-state conditions are reached when the mass flux of chloride leaving a watershed is in balance with the input of chloride to that watershed. Using the estimated

long-term average for discharge along with observed chloride inputs (2012–2019), we calculated the steady-state concentrations of chloride at the KB100 and KB150 gauges. This is given by the average rate of chloride mass added to a watershed divided by the average rate of discharge. During the short-term salt application period (2018–2019), the mean steady-state chloride concentrations for KB100 and KB150 were 133 mg/L and 207 mg/L, respectively. These concentrations were greater than the observed mean flux concentrations of 98 mg/L and 166 mg/L measured at the same gauges (S7 Fig). Our results were consistent with the observed accumulation of chloride over that period indicating that the watersheds are not yet at steady-state and are still accumulating chloride. The long-term salt application period (2012–2019) also indicated this potential for chloride accumulation: the steady-state concentrations for the entire period from 2012 to 2019 for KB100 and KB150 were 166 mg/L and 250 mg/L, respectively.

The steady-state concentrations represent the theoretical expected average concentrations at the outlets of the watersheds. Spatially, as previously indicated by Richburg et al. [26] and Rhodes and Guswa [25], a chloride concentration gradient exists in the fen region of Kampoosa Bog. Concentrations are higher closer to the Turnpike and decrease downgradient towards US-7. As a result, concentrations in the northern sections of the fen region are expected to be higher than the calculated steady-state concentration. Our study provides additional evidence for this: we found that groundwater chloride concentrations at Well A (5 ft) were higher (~200 mg/L) than the long-term (166 mg/L) and short-term (133 mg/L) steady-state concentrations in the stream measured at KB100.

The presence of the concentration gradient could impact the ecology of the fen. Past studies have documented the presence of both invasive *Phragmites* and salt tolerant *Typha angustifolia/xglauca*, which tends to diminish plant diversity, in the northern parts of the fen region [25, 26, 44]. In contrast to other studies Richburg et al. [26], Panno et al. [69], and Wilcox [70] concluded that there was no correlation between chloride concentrations and abundance of invasive *Phragmites* at Kampoosa Bog. Moreover, Richburg et al. [26] found that the overall evenness, richness, and total vegetation cover was not significantly different in areas with and those without *Phragmites*. However, in a later vegetation survey, Mays [44], showed there was a sudden increase in *Typha angustifolia/xglauca* following *Phragmites* control efforts in 2008. Increased density of *Typha angustifolia/xglauca* at Kampoosa Bog was attributed to either high salinity or disturbances from *Phragmites* control [44]. Without any reductions in road-salt application practices, increases in chloride concentrations and accumulation are expected to occur. If left untreated, increases in salt-tolerant plant species densities could be inevitable in the northern section of the fen region (as demonstrated in the past) which has higher chloride concentrations. However, even rapid reductions in road-salt applications will not result in rapid reduction in chloride concentrations due to legacy chloride in the system [71].

## Implications for road-salt application

Road-salt poses a threat to plant and aquatic communities [20]. Our study shows that heavier salt application in 2018 and 2019, coupled with the dryness of 2019 led to high chloride accumulation rates and increases in groundwater chloride concentrations. We estimate the annual chloride accumulation rates in the entire watershed to be >100000 kg/year in 2018 and 2019. Furthermore, limiting the chloride concentrations in the bog to 54 mg/L [26] and 77 mg/L– 95 mg/L has been recommended to maintain the current vegetation community. However, at the beginning of 2020, groundwater chloride concentrations in some parts of the wetland region were still greater than 250 mg/L (Fig 3).

The MassDOT has implemented Best Management Practices (BMP) to address road salt contamination in the state. Through the Snow and Ice Control Program (SICP; [72]), Mass-DOT reported a 26% decrease in salt application rates, on a kg per lane-meter basis, between 2011 and 2022. This report [72] indicates that since 2017, the department (1) improved weather forecasting and added more than 25 vehicle-mounted mobile Road Weather Information Systems (RWIS) sensors, (2) lowered average annual state salt application rates to 14 kg/lane-meter from the 2001–2010 rates of 21 kg/lane-meter, (3) improved calibration of spreader vehicles, and (4) increased the number of liquid deicer spreader vehicles (SICP). As road-salt application practices continue to improve, water quality and vegetation monitoring should also be continued to further assess the benefits of road-salt reductions to wetland ecology. Further investment in technology that improves more targeted application of deicing agents to highways to help reduce the amount of salt applied to roads will help lower the accumulation of dissolved salts to the environment.

## Conclusion

The goal of this study was to (1) investigate the effects of changes in road-salt application rates on groundwater chloride concentrations in the Kampoosa Bog drainage basin. We also (2) quantified and compared the impacts of hydrologic and salt application variations on three subwatersheds in the basin, and (3) investigated the role of wetlands in road-salt contaminated watersheds. Using mass-balance models, we calculated chloride outflux and accumulation rates in the three subwatersheds (KB300, KB150 and KB100) located in this drainage basin and used the Seasonal Mann-Kendall analysis to identify trends in the chloride accumulation rates and measured groundwater concentrations in the fen region of the watershed. From our analysis, the key findings were:

1. Increases in salt application rates have led to increases in chloride concentrations in groundwater. However, there is a delayed response time between changes in salt applications to highways and groundwater chemistry of the fen, as evidenced by low fluctuations in Cl concentrations of groundwater over time. Superimposed within the increasing trend were notable fluctuations in chloride concentrations in shallow groundwater (5ft) whereas deeper groundwater (15ft) concentrations were more consistent.

2. Despite some minor differences in the KB300 watershed, observed responses of chloride concentrations to streamflow changes were similar in all the watersheds. Streamflow dilutes chloride concentrations and increases mass flux. At the daily, monthly, and annual time-scales, chloride outflux is driven by streamflow.

3. In the Kampoosa drainage basin, wetland percent cover is not directly correlated with chloride retention efficiency, but catchments with larger wetland areas can retain more water and dissolved salts during drier years. The wetland region (which has higher percent wetland cover than the KB150 watershed) and KB150 watershed had similar retention efficiencies in 2018, but the wetland region had a higher retention in the drier year (2019). Larger watersheds with gentle slopes, such as the wetland region, tend to promote the accumulation of water while smaller watersheds, similar to KB150, that are characterized by steeper slopes, facilitate continuous surface runoff. During dry years, accumulated water in the wetland region is lost to evapotranspiration leading to lower discharge and chloride outfluxes.

4. Chloride accumulation in the Kampoosa Bog subwatersheds is still high which poses a threat to aquatic life and wetland ecology. The watersheds have not yet reached steady state conditions with respect to chloride. As a result, continued chloride accumulation and

increasing concentrations are expected if application of salt to roadways is not reduced. Implementing reductions in salt application is likely to yield significant benefits for these sensitive wetland ecosystems that are impacting plant ecology.

## Supporting information

**S1 Fig. Daily discharge-chloride mass flux relationship in the KB100, KB150 and KB300 watersheds during the four periods (SON: Sept-Oct-Nov, MAM: March-April-May, JJA: June-July-Aug, DJF: Dec-Jan-Feb).**
(TIF)

**S2 Fig. Discharge measured at the main inlet (KB300, blue line) and outlet (KB100, red) of the Kampoosa Bog drainage basin.** The grey dots indicate events when discharge is higher at the inlet during the dry months (June—Oct) during which Kampoosa Brook becomes a losing stream and recharges the fen region and groundwater.
(TIF)

**S3 Fig. Chloride concentrations normalized by lane density in the KB100 and KB150 watersheds.**
(TIF)

**S4 Fig. Chloride surface water concentrations measured in the fen region compared to the surface water at the watershed outlet (KB100).** Surface water concentrations in the fen region tend to be higher. Unlike the groundwater chemistry at the 5ft and 15ft depths, surface water does not follow a distinct spatial pattern: concentrations are not always higher at well A which is closer to I-90.
(TIF)

**S5 Fig. Monthly precipitation (mm/month) and discharge (mm/month) measured at the KB100, KB150 and KB300 gauges.** Precipitation is represented by the grey bars and the line graphs represent the different subwatersheds. High discharge is recorded during high precipitation events and in the Sept–Feb months.
(TIF)

**S6 Fig. Daily discharge-chloride flux concentration relationship in the KB100, KB150 and KB300 watersheds during the four periods (SON: Sept-Oct-Nov, MAM: March-April-May, JJA: June-July-Aug, DJF: Dec-Jan-Feb).**
(TIF)

**S7 Fig. Daily chloride mass flux normalized by lane length.** The mean and median mass fluxes are calculated.
(TIF)

**S8 Fig. Monthly water storage in the KB100 subwatershed.** An arbitrary initial storage value of 800000 $m^3$ was used in this study. Changes in water storage follow a seasonal pattern and water storage tends to be higher in the Sept—Feb months.
(TIF)

**S9 Fig. Low chloride concentrations from monthly grab samples at both MB-100 and KB-300 when compared to KB-100 and KB-150.** Concentrations at MB-100 are considered to be the background concentrations independent of road-salt added to I-90 and US-7.
(TIF)

**S1 Table. Historic chloride application rates in the Kampoosa Bog subwatersheds between 2012 and 2019.**
(DOCX)

## Acknowledgments

We would like to thank Erich Hinlein and Camelia Rotaru of The University of Massachusetts-Amherst for collecting and analyzing the field samples and curating the streamflow data used in this study. We also thank the Massachusetts Department of Transportation (MassDOT) and Laurene Poland from MassDOT for providing the road salt application data.

## Author Contributions

**Conceptualization:** Wayne Ndlovu, Andrew J. Guswa, Amy L. Rhodes.

**Formal analysis:** Wayne Ndlovu, Andrew J. Guswa, Amy L. Rhodes.

**Funding acquisition:** Amy L. Rhodes.

**Methodology:** Wayne Ndlovu, Andrew J. Guswa, Amy L. Rhodes.

**Supervision:** Andrew J. Guswa, Amy L. Rhodes.

**Validation:** Wayne Ndlovu, Andrew J. Guswa, Amy L. Rhodes.

**Visualization:** Wayne Ndlovu, Andrew J. Guswa, Amy L. Rhodes.

**Writing – original draft:** Wayne Ndlovu.

**Writing – review & editing:** Andrew J. Guswa, Amy L. Rhodes.

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
