## [Decision Letter · Decision Letter 0]

16 Jul 2024

PONE-D-24-23271Accumulation of road salt in a calcareous fen: Kampoosa Bog, western MassachusettsPLOS ONE

Dear Dr. Ndlovu,

Thank you for submitting your manuscript to PLOS ONE. After careful consideration, we feel that it has merit but does not fully meet PLOS ONE’s publication criteria as it currently stands. Therefore, we invite you to submit a revised version of the manuscript that addresses the points raised during the review process.

We look forward to receiving your revised manuscript.

Kind regards,

Deepak Singh, Ph D

Academic Editor

PLOS ONE

Journal Requirements:

Please ensure that your manuscript meets PLOS ONE's style requirements, including those for file naming. The PLOS ONE style templates can be found at https://journals.plos.org/plosone/s/file?id=wjVg/PLOSOne_formatting_sample_main_body.pdf and https://journals.plos.org/plosone/s/file?id=ba62/PLOSOne_formatting_sample_title_authors_affiliations.pdf 2. We note that the grant information you provided in the ‘Funding Information’ and ‘Financial Disclosure’ sections do not match.  When you resubmit, please ensure that you provide the correct grant numbers for the awards you received for your study in the ‘Funding Information’ section. 3. Thank you for stating the following financial disclosure: Wayne Ndlovu reports financial support was provided by Massachusetts Division of Fisheries and Wildlife. Amy L. Rhodes reports financial support was provided by Massachusetts Division of Fisheries and Wildlife. If there are other authors, they declare that they have no known competing financial interests or personal relationships that could have appeared to influence the work reported in this paper. Please state what role the funders took in the study.  If the funders had no role, please state: "The funders had no role in study design, data collection and analysis, decision to publish, or preparation of the manuscript." If this statement is not correct you must amend it as needed. Please include this amended Role of Funder statement in your cover letter; we will change the online submission form on your behalf. 4. Thank you for stating the following in the Acknowledgments Section of your manuscript: This work was supported by funding from the Natural Heritage and Endangered Species Program of the Massachusetts Division of Fisheries & Wildlife in the Department of Fish and Game (RFR #DFW-2020-051), McKinley Fellowship and the Tomlinson Fund (Smith College). We would like to thank Erich Hinlein and Camelia Rotaru of The University of Massachusetts-Amherst for collecting and analyzing the field samples and curating the streamflow data used in this study. We also thank the Massachusetts Department of Transportation (MassDOT) and Laurene Poland from MassDOT for providing the road salt application data. We note that you have provided funding information that is not currently declared in your Funding Statement. However, funding information should not appear in the Acknowledgments section or other areas of your manuscript. We will only publish funding information present in the Funding Statement section of the online submission form. Please remove any funding-related text from the manuscript and let us know how you would like to update your Funding Statement. Currently, your Funding Statement reads as follows: Wayne Ndlovu reports financial support was provided by Massachusetts Division of Fisheries and Wildlife. Amy L. Rhodes reports financial support was provided by Massachusetts Division of Fisheries and Wildlife. If there are other authors, they declare that they have no known competing financial interests or personal relationships that could have appeared to influence the work reported in this paper.
 Please include your amended statements within your cover letter; we will change the online submission form on your behalf. 5. Thank you for uploading your study's underlying data set. Unfortunately, the repository you have noted in your Data Availability statement does not qualify as an acceptable data repository according to PLOS's standards. At this time, please upload the minimal data set necessary to replicate your study's findings to a stable, public repository (such as figshare or Dryad) and provide us with the relevant URLs, DOIs, or accession numbers that may be used to access these data. For a list of recommended repositories and additional information on PLOS standards for data deposition, please see https://journals.plos.org/plosone/s/recommended-repositories. 6. Your ethics statement should only appear in the Methods section of your manuscript. If your ethics statement is written in any section besides the Methods, please delete it from any other section.  7. We notice that your supplementary tables are included in the manuscript file. Please remove them and upload them with the file type 'Supporting Information'. Please ensure that each Supporting Information file has a legend listed in the manuscript after the references list. 8. Please review your reference list to ensure that it is complete and correct. If you have cited papers that have been retracted, please include the rationale for doing so in the manuscript text, or remove these references and replace them with relevant current references. Any changes to the reference list should be mentioned in the rebuttal letter that accompanies your revised manuscript. If you need to cite a retracted article, indicate the article’s retracted status in the References list and also include a citation and full reference for the retraction notice.

Additional Editor Comments:

Dear Wayne Ndlovu,

Your manuscript entitled:

"Accumulation of road salt in a calcareous fen: Kampoosa Bog, western Massachusetts" has been reviewed. The reviewer(s) have recommended minor revisions for your manuscript and the comments are included at the bottom of this letter.

Sincerely,

Dr. Deepak Singh

Academic Editor

PLOS ONE

Reviewers' comments:

Reviewer's Responses to Questions

**Comments to the Author**

1. Is the manuscript technically sound, and do the data support the conclusions?

Reviewer #1: Yes

Reviewer #2: Yes

2. Has the statistical analysis been performed appropriately and rigorously? 

Reviewer #1: Yes

Reviewer #2: Yes

3. Have the authors made all data underlying the findings in their manuscript fully available?

Reviewer #1: Yes

Reviewer #2: Yes

4. Is the manuscript presented in an intelligible fashion and written in standard English?

Reviewer #1: Yes

Reviewer #2: Yes

5. Review Comments to the Author

**Reviewer #1: **This manuscript provides a thorough and well-researched analysis of the impacts of road salt on a calcareous fen. The study's findings are significant for understanding the long-term ecological impacts of road salt and informing management practices. Minor improvements in the figures and more detailed descriptions in the methods section would enhance the manuscript’s quality.

**Reviewer #2:** The authors must address the comments in the manuscript especially, the major limitations of the study based on the practical problems faced during the period of data collection. Authors also need to justify why the Chloride Retention efficiency is more in KB 300 during 2018 and 2019 even without having any wetland area and why the chloride flux in the KB300 was lower than the other two, though the Chloride Retention efficiency was more during 2018 and 2019.

6. PLOS authors have the option to publish the peer review history of their article (what does this mean?). If published, this will include your full peer review and any attached files.

Reviewer #1: No

Reviewer #2: No

---

## [Author Response · Author response to Decision Letter 0]

18 Sep 2024

Response to reviewer’s comments

Thank you for taking the time to provide us with feedback on how we can improve our manuscript. Our responses to the reviewers are as follows:

Academic Editor’s Comments:

Response:

Thank you for providing these style templates. To address the formatting issues, we have done the following:

 Formatted the title page to exclude the zip codes from the affiliations and instead included the city, state and country.

 Updated the affiliations bullet style in the title page.

 Deleted the ‘Keywords’ section from the manuscript.

 Edited all headings and subheadings to follow sentence case style and updated their font sizes.

 Edited the figure captions to highlight the figure numbers and titles by making them bold.

 Updated the referencing of supplementary figures.

 Made the table titles bold.

 Deleted the funding information from the Acknowledgement section

 Updated the reference style to the Vancouver style.

Response:

We have updated the Funding Information section to reflect the correct grant information.

Wayne Ndlovu reports financial support was provided by Massachusetts Division of Fisheries and Wildlife. Amy L. Rhodes reports financial support was provided by Massachusetts Division of Fisheries and Wildlife. If there are other authors, they declare that they have no known competing financial interests or personal relationships that could have appeared to influence the work reported in this paper.

Response:

We have included the Role of Funder statement in the cover letter.

This work was supported by funding from the Natural Heritage and Endangered Species Program of the Massachusetts Division of Fisheries & Wildlife in the Department of Fish and Game (RFR #DFW-2020-051), McKinley Fellowship and the Tomlinson Fund (Smith College). We would like to thank Erich Hinlein and Camelia Rotaru of The University of Massachusetts-Amherst for collecting and analyzing the field samples and curating the streamflow data used in this study. We also thank the Massachusetts Department of Transportation (MassDOT) and Laurene Poland from MassDOT for providing the road salt application data.

Wayne Ndlovu reports financial support was provided by Massachusetts Division of Fisheries and Wildlife. Amy L. Rhodes reports financial support was provided by Massachusetts Division of Fisheries and Wildlife. If there are other authors, they declare that they have no known competing financial interests or personal relationships that could have appeared to influence the work reported in this paper.

Response:

We removed all funding related information from the manuscript and added the amended statement within the cover letter.

5. Thank you for uploading your study's underlying data set. Unfortunately, the repository you have noted in your Data Availability statement does not qualify as an acceptable data repository according to PLOS's standards.

Response:

We have published our data and related code to HydroShare and below are the citation and URL:

Ndlovu, W., A. Guswa, A. L. Rhodes (2024). Kampoosa Bog - Road salt accumulation, HydroShare, http://www.hydroshare.org/resource/2c676a5913f44c85befedfcedc6185bc

Response:

We moved the ethics statements to the Methods section (Lines 241-244).

7. We notice that your supplementary tables are included in the manuscript file. Please remove them and upload them with the file type 'Supporting Information'. Please ensure that each Supporting Information file has a legend listed in the manuscript after the references list.

Response:

We deleted the supplementary tables that were included in the manuscript and included them in the Supporting Information file.

Response:

We thank the editor for the comment. We did not include any retracted articles in our manuscript, however, some of our citations had not been included in the references. We have relinked our citations and included the following references in the revised manuscript:

3. Pieper KJ, Tang M, Jones CN, Weiss S, Greene A, Mohsin H, et al. Impact of Road Salt on Drinking Water Quality and Infrastructure Corrosion in Private Wells. Environ Sci Technol. 2018 Dec 18;52(24):14078–87. 

21. US EPA O. Secondary Drinking Water Standards: Guidance for Nuisance Chemicals [Internet]. 2015 [cited 2023 Feb 15]. Available from: https://www.epa.gov/sdwa/secondary-drinking-water-standards-guidance-nuisance-chemicals

41. Kampoosa Bog Drainage Basin ACEC | Mass.gov [Internet]. [cited 2022 Apr 10]. Available from: https://www.mass.gov/service-details/kampoosa-bog-drainage-basin-acec

49. Rhodes A, Wetzel P, Ndlovu W. Kampoosa Bog Water Quality and Vegetation Composition Review and Analysis. 2021. 

50. Latham, NY Weather History | Weather Underground [Internet]. 2023 [cited 2023 Sep 28]. Available from: https://www.wunderground.com/history/monthly/us/ny/latham/KALB/date/2019-12

51. CoCoRaHS - Community Collaborative Rain, Hail & Snow Network [Internet]. [cited 2022 Mar 14]. Available from: https://www.cocorahs.org/

63. August 2018 National Climate Report | National Centers for Environmental Information (NCEI) [Internet]. [cited 2023 Jul 31]. Available from: https://www.ncei.noaa.gov/access/monitoring/monthly-report/national/201808

72. Mass DOT Snow And Ice Control Program 2022 Environmental Status And Planning Report [Internet]. Massachusetts Department of Transportation; 2022. Report No.: EEA# 11202. Available from: https://archive.org/details/mass-dot-snow-and-ice-control-program-2022-environmental-status-and-planning-report-eea-11202_202306/page/n25/mode/2up

Ronalyn M. Ramos Comments

1. We note that Figure 1 in your submission contain map/satellite images which may be copyrighted. All PLOS content is published under the Creative Commons Attribution License (CC BY 4.0), which means that the manuscript, images, and Supporting Information files will be freely available online, and any third party is permitted to access, download, copy, distribute, and use these materials in any way, even commercially, with proper attribution. For these reasons, we cannot publish previously copyrighted maps or satellite images created using proprietary data, such as Google software (Google Maps, Street View, and Earth). For more information, see our copyright guidelines: https://nam10.safelinks.protection.outlook.com/?url=http%3A%2F%2Fjournals.plos.org%2Fplosone%2Fs%2Flicenses-and-copyright&data=05%7C02%7Cwndlovu%40ku.edu%7C0c8083b78e5c4618822208dcd3ddd5cb%7C3c176536afe643f5b96636feabbe3c1a%7C0%7C0%7C638618196958796246%7CUnknown%7CTWFpbGZsb3d8eyJWIjoiMC4wLjAwMDAiLCJQIjoiV2luMzIiLCJBTiI6Ik1haWwiLCJXVCI6Mn0%3D%7C0%7C%7C%7C&sdata=wpcTkybCcS58MVQyCtXPQVHfHnYtItHuA5fbuCFidW4%3D&reserved=0.

Response:

We thank the reviewer for correction. We have created a new map for Figure 1 which does not contain contain copyrighted maps from Google. The new map was created using ESRI ArcPro and using public domain aerial photography and ESRI digital elevation models. We also edited the figure caption for Figure 1 and attributed the data sources and software used.

Reviewer comments (Page, Line)

Reviewer #2:

(2, 28) Suggestion to change from water sources to water resources.

Response:

We accepted the reviewer’s suggestion, and the sentence now reads as:

“Road salt poses a threat to the quality of soils and water resources.”

(4, 99) Suggestion to delete ‘this’.

Response:

We thank the reviewer for the correction and have accepted the suggestion.

(5, 134) Suggestion to replace adsorprtion with adsorption.

Response:

We thank the reviewer for highlighting the spelling error. We accepted the reviewer’s suggestion and the sentence now reads as:

“Over the years, constructed wetlands have been used as a less-expensive alternative to store and treat stormwater and wastewater via adsorption and diffusion into pores [31–37].”

(5, 139-140) Suggestion to add a comma and the word ‘the’.

Response:

We accepted the reviewer’s corrections, and the sentence now reads as follows:

“Two major highways cross the watershed, Interstate-90 (Massachusetts Turnpike), and U.S. Route 7 (US-7) and are the major sources of road salt every winter.”

(6, 175-177) Suggestion to correct the spelling for ‘additionally’ and the structure of the sentence.

Response

We thank the reviewer for the suggestions and have edited the sentence as follows:

“Additionally, increase in sodium leads to decreases in biomass accumulation, inhibits plant growth and promotes crop senescence [47,48].”

(8, 275) The coordinates must be mentioned properly

Response:

We thank the reviewer for the correction and have corrected the coordinates in line 192:

“The Kampoosa Bog drainage basin (Fig 1) is a 465 ha watershed located in western Massachusetts (42.293 N, 73.305W) that comprises ponds, a graminoid fen, shrub fens and red maple swamps [25,26].”

(12, 400) Added ‘Eq’ to equation reference

(15, 491) Added ‘Eq’ to equation reference

(16, 510) Added ‘Eq’ to equation reference

(16, 516) Added ‘Eq’ to equation reference

(17, 539) Added ‘Eq’ to equation reference

Response:

We thank the reviewer for the corrections and have accepted the suggestions.

[11, 342 – 359] What are the major limitations of the study based on the practical problems faced during the period of data collection.

Response

Thank you for this comment. We first edited a typing error which stated that we included ‘irregular data collected at KB150 caused by ice formation during Dec 2018’ (Page 11, Line 375) in our analysis and corrected it to show that the ice formation was during Dec 2017, a period excluded from this study. We also added an additional sentence at the end of this paragraph describing a potential limitation of interpolating some of the data. The paragraph now reads as follows:

“At the four gauges, water level and temperature (oC) plus specific conductance (μS/cm) were measured every fifteen minutes from Nov 2017 to Oct 2020 using a HOBO U20-001-04 water level data logger) and a HOBO U24-001 conductivity data logger (Onset Computer Corporation; Bourne, MA), respectively. However, data from the MB100 gauge was excluded from the analysis due to conductivity probe malfunctions during this period. We focus on data collected from Jan 2018 to April 2020 due to (1) irregular data collected at KB150 caused by ice formation during Dec 2017 and (2) the presence of a beaver dam at the KB100 culvert from May 2020 to Oct 2020 which flooded the fen region and impacted the stage-discharge relationship. Other irregularities in the data caused by stream freeze-thaw cycles occurred over shorter time frames ranging from a few hours to less than 10 days and were addressed by interpolating between reliable data points. Interpolating between these data points may have failed to fully account for some of the complex variations in stage and discharge, which could have led to an oversimplified representation of the data and potentially obscured other short-term fluctuations, but we believe these to be minor given the short timeframe.”

(19, 600-601) Suggestion to us past tense ‘indicated’ and to add the word ‘to’

Response:

We thank the reviewer for the corrections and have accepted the changes. The revised sentence now reads as:

“Historic road salt application data indicated that salt application rates were slightly higher in 2018 and 2019 compared to 2012 – 2017 (Fig 4, S1 Table).”

(Page 20) Table 2. Kindly justify why the Chloride Retention efficiency is more in KB 300 during 2018 and 2019 even without having any wetland area as per Table 1.

Response:

The delineation of the portion of the Turnpike that contributes road salt to KB300 is challenging and we recognize that this could have introduced a degree of uncertainty to our calculations. Our delineation method could have led to overestimating the road length that is contributing road salt to KB300 which led to overestimating the amount of chloride retained in the subwatershed. To highlight this, we added the following: 

“Additional uncertainty to these data could also be introduced by the uncertainty associated with estimating the road length in each subwatershed that is contributing salt.” (14, 462-464)

“Although the KB300 subwatershed has no known wetlands, we estimated relatively high retention rates during 2018 and 2019. This could be due to the uncertainty introduced during the watershed delineation resulting in the overestimation of the road length contributing road salt to the KB300 subwatershed.” (19, 618-621)

(24, 739-745) Why the chloride flux in the KB300 was lower than the other two, though the Chloride Retention efficiency is more during 2018 and 2019

Response:

We calculated chloride flux concentration as the total mass of chloride per total volume of water over a period, as stated in line 718-719 while chloride retention is calculated using Eq 4 as shown below:

Ann

---

## [Decision Letter · Decision Letter 1]

4 Oct 2024

Accumulation of road salt in a calcareous fen: Kampoosa Bog, western Massachusetts

PONE-D-24-23271R1

Dear Dr. Ndlovu,

We’re pleased to inform you that your manuscript has been judged scientifically suitable for publication and will be formally accepted for publication once it meets all outstanding technical requirements.

Kind regards,

AL MAHFOODH

Academic Editor

PLOS ONE

Additional Editor Comments (optional):

Reviewers' comments:

Reviewer's Responses to Questions

**Comments to the Author**

1. If the authors have adequately addressed your comments raised in a previous round of review and you feel that this manuscript is now acceptable for publication, you may indicate that here to bypass the “Comments to the Author” section, enter your conflict of interest statement in the “Confidential to Editor” section, and submit your "Accept" recommendation.

Reviewer #1: All comments have been addressed

Reviewer #2: All comments have been addressed

2. Is the manuscript technically sound, and do the data support the conclusions?

Reviewer #1: Yes

Reviewer #2: Yes

3. Has the statistical analysis been performed appropriately and rigorously? 

Reviewer #1: Yes

Reviewer #2: Yes

4. Have the authors made all data underlying the findings in their manuscript fully available?

Reviewer #1: Yes

Reviewer #2: Yes

5. Is the manuscript presented in an intelligible fashion and written in standard English?

Reviewer #1: Yes

Reviewer #2: Yes

6. Review Comments to the Author

Reviewer #1: This manuscript provides a thorough and well-researched analysis of the impacts of road salt on a calcareous fen. The study's findings are significant for understanding the long-term ecological impacts of road salt and informing management practices.

Reviewer #2: The authors have addressed all the comments. The effort taken by the authors to repond to every comment and the justification given is commendable.

7. PLOS authors have the option to publish the peer review history of their article (what does this mean?). If published, this will include your full peer review and any attached files.

Reviewer #1: No

Reviewer #2: No

---

## [Editor Report · Acceptance letter]

11 Oct 2024

PONE-D-24-23271R1 

PLOS ONE

Dear Dr. Ndlovu, 

I'm pleased to inform you that your manuscript has been deemed suitable for publication in PLOS ONE. Congratulations! Your manuscript is now being handed over to our production team.

Kind regards, 

on behalf of

Dr. AL MAHFOODH 

Academic Editor

PLOS ONE